# Thermogenesis and Energy Metabolism in Brown Adipose Tissue in Animals Experiencing Cold Stress

**DOI:** 10.3390/ijms26073233

**Published:** 2025-03-31

**Authors:** Xuekai Zhang, Jin Xiao, Min Jiang, Clive J. C. Phillips, Binlin Shi

**Affiliations:** 1College of Animal Science, Inner Mongolia Agricultural University, Hohhot 010018, China; zxk1101101102023@163.com (X.Z.); jiangmin080808@163.com (M.J.); shibinlin@yeah.net (B.S.); 2Curtin University Sustainability Policy (CUSP) Institute, Curtin University, Perth, WA 6845, Australia; clive.phillips@curtin.edu.au; 3Institute of Veterinary Medicine and Animal Sciences, Estonian University of Life Sciences, 51006 Tartu, Estonia

**Keywords:** cold stress, brown adipose tissue, non-shivering thermogenesis, mitochondria homeostasis, energy metabolism and balance

## Abstract

Cold exposure is a regulatory biological functions in animals. The interaction of thermogenesis and energy metabolism in brown adipose tissue (BAT) is important for metabolic regulation in cold stress. Brown adipocytes (BAs) produce uncoupling protein 1 (UCP1) in mitochondria, activating non-shivering thermogenesis (NST) by uncoupling fuel combustion from ATP production in response to cold stimuli. To elucidate the mechanisms underlying thermogenesis and energy metabolism in BAT under cold stress, we explored how cold exposure triggers the activation of BAT thermogenesis and regulates overall energy metabolism. First, we briefly outline the precursor composition and function of BA. Second, we explore the roles of the cAMP- protein kinase A (PKA) and adenosine monophosphate-activated protein kinase (AMPK) signaling pathways in thermogenesis and energy metabolism in BA during cold stress. Then, we analyze the mechanism by which BA regulates mitochondria homeostasis and energy balance during cold stress. This research reveals potential therapeutic targets, such as PKA, AMPK, UCP1 and PGC-1α, which can be used to develop innovative strategies for treating metabolic diseases. Furthermore, it provides theoretical support for optimizing cold stress response strategies, including the pharmacological activation of BAT and the genetic modulation of thermogenic pathways, to improve energy homeostasis in livestock.

## 1. Introduction

Extreme environmental temperature affects the health, welfare, and productivity of animals and in some circumstances results in death [1]. Links between environmental temperature and animal production have attracted the attention of the farming community and researchers [2]. Both high and low temperatures cause thermal stress that limits animal production and welfare. Environmental-temperature-related mortality is primarily linked to cold environments [3]. Prolonged exposure to extreme cold prompts physiological and psychological responses to maintain body temperature, which include vasoconstriction, muscle shivering, and heightened metabolism, to safeguard against cold stress [4]. Health issues such as cold tremors, frostbite, and hypothermia may ensue when livestock is challenged by extreme cold conditions. These disrupt energy metabolism, neuroendocrine function, and cardiovascular and reproductive systems and if untreated may eventually lead to death [5]. If feed is readily available, continuous exposure to cold environments increases animals’ energy intake to support the maintenance of body temperature, boosting internal thermogenesis. However, this reduces feed efficiency and productivity, thereby restricting the economic growth of the livestock industry [6]. Total energy expenditure is increased, leading to metabolic adaptations, increased feed intake, liver glucose production, and glucose utilization by adipose tissue [7]. Hence, the thermogenic dimension of energy homeostasis in animals is of interest to develop methods to overcome cold stress.

White adipose tissue (WAT) provides an energy store and is involved in obesity. Brown adipose tissue (BAT) was first identified by Conrad Gessner in 1551, but it was only in 1961 that it was firmly identified as a thermogenic organ [8]. BAT is more common in newborns and young mammals. BAT regulates energy metabolism through indirect thermogenesis, and with age, it gradually transitions to white adipocytes [9]. Cold stimulation increases norepinephrine (NE) by activating non-shivering thermogenesis (NST). Increased NE upregulates peroxisome-proliferator-activated receptor gamma coactivator-1α (PGC-1α), which activates the specific uncoupling protein 1 (UCP1) on the inner membrane of brown adipose (BA) mitochondria to generate heat [10]. At the same time, oxidative phosphorylation in mitochondria is diminished through the action of UCP1 without producing ATP, leading to the dispersal of the energy produced by mitochondria in the form of heat [10]. BAT is essential for energy metabolism, stimulating NST through uncoupled oxidative phosphorylation (OXPHOS) to activate the mitochondrial respiratory chain, which stimulates thermogenesis [11]. BAT is a specialized for NST, and its effects on energy metabolism are independent of energy expenditure and body conformation regulation [12]. UCP1 in BA enables the oxidation of glucose and fatty acids, converting food energy into heat to resist the cold [13].

The thermogenesis and energy metabolism processes in BA during cold stress involve multiple signaling pathways, such as cAMP- protein kinase A (PKA) and adenosine monophosphate-activated protein kinase (AMPK) [14]. The activation of BA to reduce adipose tissue accumulation has implications for the prevention of obesity, associated metabolic disorders, and mammalian adaptation to cold exposure [14]. Cold exposure is a major cause of mortality in lambs, especially when lambs are born in late winter [15]. Prior to parturition, placental PGE2 prevents brown fat from thermogenesis, but after birth, the differential between the ewe’s thermal environment and that of her lamb(s) is crucial in determining whether lambs can survive by mobilizing BAT. Early parturition in a fully fleeced pregnant ewe stimulated by warm conditions leads to reduced BAT in lambs and reduced ability to thermoregulate in cold conditions. Compared with rearing by the ewe, the hand-rearing of lambs retards the transition between BAT and WAT, delaying postnatal development [16,17].

In this review, we discuss the precursor composition and function of BA and its role in the regulation of thermogenesis and energy homeostasis under cold stress.

## 2. Precursor Composition and Function of Brown Adipocytes

Adipose tissue, commonly referred to as “fat”, is a passive energy reservoir, participating in physiological processes, such as regulating food intake, energy maintenance, insulin secretion, immune response, and body temperature regulation [18,19]. There are three distinct adipocyte types in mammals: white adipocytes, BA, and beige adipocytes [20,21]. WAT is widely distributed in subcutaneous tissue and around viscera. Its cell morphology includes a single large lipid droplet, with few mitochondria and little cytoplasm, and it acts as an energy store. WAT is also an endocrine organ, secreting adipokines such as leptin, lipotropin, and tumor necrosis factor α, regulating energy metabolism and maintaining physiological homeostasis [22]. Although BAT is more common in neonates and young mammals, some functional BAT exists in adults, located near the vertebrae and clavicle in the form of a brown butterfly between the shoulder blade, in the back of the neck, in the mediastinum and around the kidneys. The cell size is small, with cells containing many small lipid droplets and being rich in mitochondria and cytochromes. The cell surface is densely covered with sympathetic nerve fibers and capillaries [12,23]. In cold stress, the brown-fat-defining marker protein UCP1 is activated to increase thermogenesis and energy expenditure via NST, unlike in WAT [10]. Beige adipocytes have some lipid droplets and mitochondria, which have a similar function to BA but are weakly expressed [21].

There are two types of BAs in mammals: classical BAs in the interscapular region of rodents and BAs from brown precursor adipocytes [24]. The browning of adipocytes and beige adipocytes occurs from white adiposes or adipose-tissue-derived stem cells [25] (Table 1).

Myogenic factor 5 (Myf-5) is a regulator. The precursor development of BA is similar to skeletal muscle cells that express Myf-5 [26] and different from WAT depots, which do not express Myf-5 [27]. BA is differentiated in vivo from the Myf-5-positive myoblast lineage through the action of PR domain-containing protein 16 (PRDM16) [28]. Thus, Myf-5 precursor cells can differentiate into BA or skeletal muscle cells. PMDR16 is a molecular switch for BA formation, affecting the expression of the thermogenic genes UCP1, PGC-1α, and nuclear respiratory factor 1 (Nrf1) in BA [27]. The CCAAT/enhancer binding protein β (C/EBPβ) binds to PRDM16 to induce the transformation into BA of both mouse and human skin fibroblasts. UCP1 expression is reduced, while skeletal-muscle-specific gene expression is increased; thereby, brown adipocyte differentiation capacity is weakened in the absence of C/EBPβ [28]. Furthermore, bone morphogenetic protein (BMP) has a pivotal role in brown adipose differentiation. Unlike other regulators, BMP not only promotes the differentiation of brown adipose precursor cells but also significantly reduces intracellular lipid accumulation and induces UCP1 expression [29]. BMP-7 stimulates BA by inducing PRDM16 and PGC-1α and increasing the expression of BA-defining marker UCP1, the adipogenesis transcription factor peroxisome-proliferator-activated receptor γ (PPARγ) and C/EBP [30]. BMP-7 also induces BA mitochondrial biogenesis by p38 mitogen-activated protein kinase (MAPK) activation of PGC-1α to promote the differentiation of brown adipose precursor cells [31]. BMP7 knockout (KO) mice embryos with little BAT express very little UCP1, and the interscapular BAT content of neonatal BMP7 KO mice was markedly reduced to only 30~50% of that in wild mice [29]. Conversely, BMP7-overexpressing mice showed significantly increased BA content and energy consumption but reduced weight gain.

These studies suggest that although there are different sources of BA formation, they collectively regulate the expression of UCP1 in BA. Thus, BA has both thermogenesis and energy-regulating effects in precursor development [32].

**Table 1 ijms-26-03233-t001:** The characteristics of brown, beige, and white adipocytes.

Item	Brown Adipocytes	Beige Adipocytes	White Adipocytes	References
	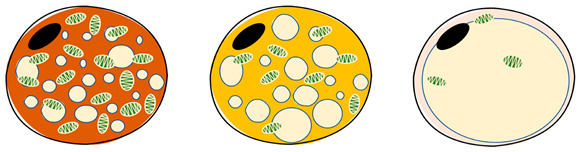			[33,34]
Location	Interscapular (young animals), supraclavicular, dorsal neck, mediastinum and around kidneys	Supraclavicular, predominantly dispersed in white and brown adipose tissue	Subcutaneous and visceral around	
Morphology	Butterfly-shaped, multilocular adipose cell	Spherical, peasized, multilocular adipose cell	Spherical, single vesicular fat cell	
Color	Brown	Beige	White	
Proportion of mitochondria	High	Medium	Low	
Lipid drops	Multilocular, small	Multilocular, small	Unilocular, occupying approximately the entire cell	
UCP1 expression	High	Medium	Low/undetectable	
Function	Consume energy (triglyceride storage), thermogenesis (non-shivering thermogenesis)	Thermogenic potential	Storing energy	
Thermogenetic activity	High	Medium	Low	
Role	endocrine organ, energy store	endocrine organ, energy store	endocrine organ, energy store, insulation	

## 3. Thermogenesis and Energy Metabolism in Brown Adipose Tissue Under Cold Stress

BA contains abundant mitochondria and induces changes in the uncoupled mitochondria proton gradient based on NST during cold stress, which regulates the thermogenesis and energy balance by dissipating energy in the shape of converted heat by highly expressed UCP1 [35]. BAT is an important organ controlling energy dissipation in rodents, and as described above, it activates UCP1 to mediate uncoupled mitochondria respiration [36]. Cypess et al. [37] showed that cold exposure for just 1 h could activate the uptake and utilization of metabolic fuels in BAT, with energy expenditure increased by an average of 79 kcal/d per 15 mL. Energy consumption in humans exposed to an acute cold environment of −19~−16 °C was greater than in those exposed to a room temperature of 24 °C, which depended on NST in BAT [38]. The cold stimulation of BAT makes it serve as an energetically and metabolically beneficial organ that converts chemical energy into heat, thereby increasing energy expenditure and maintaining energy homeostasis in the body.

The promotion of BA proliferation and differentiation by cold stress, and associated UCP1 expression, generates heat to maintain body temperature. This process utilizes multiple mechanisms, including sympathetic nerve activation, hormone release, and enhanced mitochondria activity [39]. Cold stress stimulates the sympathetic nervous system (SNS), and the NE released by the SNS promotes fat breakdown [40]. BAT helps protect against hypothermia in cold stress through adaptive thermogenesis, which is essential to maintain the whole-body energy balance [41]. When animals live in a cold environment for prolonged periods, cold stimuli induce signals, which are transmitted to the brain via cold-activated temperature sensors and transient receptor potential (TRP) on the surface sensory neurons of the body, which activates the SNS [22]. The SNS plays an important role in controlling NST in rodents and humans [42]. After β3-adrenergic receptors (β3-ARs) are activated, NE signaling increases glucose uptake in BA in vitro and in vivo. When the SNS is activated, NE is released, which stimulates BA via β3-AR. β3-AR activation induces the expression of PGC-1α in the cAMP-PKA signaling pathway and the AMPK signaling pathway, inducing BA thermogenesis [43,44]. In addition, sympathetic signaling directly activates the AMPK pathway through the adrenergic receptor α1A (ADRA1A) on the cell membrane surface to regulate energy homeostasis [45]. Cold stress triggers the release of the hormones inside cells through the SNS, which promotes triglyceride hydrolysis and fatty acid oxidation. As these metabolic pathways are activated, the expression and function of UCP1 in BA are enhanced, resulting in thermogenesis [46]. The activation of BA thermogenic protein UCP1 is coregulated by exposure to cold or NE [47]. Thus, thermogenesis and the energy metabolism processes of BAT are regulated by cold stress and SNS through the cAMP-PKA and AMPK signaling pathways to generate metabolic heat in response to cold stimuli.

### 3.1. Regulation of BAT Thermogenesis in Cold Exposure by the cAMP-PKA Pathway

#### 3.1.1. PKA

PKA, or cyclic-AMP dependent protein kinase A, comprises two catalytic subunits attached to regulatory subunits [48]. cAMP is a key regulator of lipolysis in white adipocytes and BA, mediated through PKA [49]. The binding of cAMP to PKA induces holoenzyme dissociation and catalytic subunit release and triggers the phosphorylation of PKA substrates [50]. The PKA substrate phosphorylation is critical for a variety of cellular functions, including metabolism, differentiation, signal transduction, ion channel activity, growth, and development [51]. In addition, PKA promotes free fatty acid (FFA) release to increase thermogenesis. Therefore, the PKA signaling pathway within BA can lead to stable alterations in thermogenesis, as well as in energy storage and utilization.

#### 3.1.2. PKA Pathway Activation Induces BA Thermogenesis

Under cold stress, the principal hormones released are epinephrine and noradrenaline, directly or indirectly via the hypothalamic/sympathetic nervous system, activating G-protein-coupled adenylate cyclase (AC) via the membrane receptor β3-AR to increase cAMP [52,53]. It is known that the β3-AR agonists activate BA in both rats and humans through the PKA pathway that triggers the rise in cAMP during cold stress [54].

PKA phosphorylates nuclear factor CREBs during cold stress, leading to UCP1 gene transcription in BA [55,56]. The downstream gene of the cAMP response-origin binding protein (CREB), zinc finger protein (Zfp516), joins with demethylase 1 (LSD1) and PRDM16. Both PRDM16 and LSD1 are recruited for UCP1 activation, supporting BA thermogenesis [57]. The cAMP-dependent transcription process of UCP1 is regulated by p38MAPK, an essential step in UCP1 gene transcription in mice [58,59]. β3-AR and PKA are activated under cold exposure, leading to the highly selective activation of the p38α isoform of MAPK, which in turn promotes UCP1 gene transcription [55]. The PKA-stimulated BA thermogenesis depends on UCP1 gene regulation by the downstream substrate transcription factor 2 (ATF-2) of PGC-1α in cold stress [59]. The activation of p38MAPK phosphorylates PGC-1α directly, thereby increasing UCP1 expression. PGC-1α is a transcriptional coactivator and mediator of mitochondria formation, inducing BA mitochondrial biogenesis [60]. Fisher et al. [61] showed that thermogenesis in both differentiated brown and inguinal adipocytes necessitates the presence of PGC-1α, as evidenced by their findings in a PGC-1α KO model. p38 MAPK activates PGC-1α, which enhances its activity of PGC-1α as a gene transcriptional coactivator. p38 MAPK coordinates with its downstream factor PPARγ, which binds to the UCP1 peroxisome-proliferator-activated receptor (PPAR) response element (PPRE) promoter to activate PGC-1α to induce UCP1 transcription [62]. The p38MAPK was activated in the nucleus to phosphorylate ATF-2, the downstream complex substrates Zfp516 and PRDM16 of CREB, promoting the transcription of PGC-1α, to then activate UCP1 in BA [63,64]. Thus, p38 MAPK is an important bridge in the induction of PKA on UCP1 activity in BA thermogenesis. 

In addition, the PKA signaling pathway promotes lipolysis. Non-esterified fatty acids (NEFAs) are energy-dense substrates of BA thermogenesis. The cAMP activates PKA, which phosphorylates hormone-sensitive lipase (HSL), adipose triglyceride lipase (ATGL), and the outer mitochondria membrane to initiate lipolysis, releasing FFA for utilization in mitochondria to regulate BA thermogenesis [65,66]. Nevertheless, ATGL KO results in a reduction in cAMP-mediated fatty acid oxidation and oxidative phosphorylation-related genes to reduce BA thermogenesis [67]. Perilipin (PLIN), a scaffold found on the surface of lipid droplets, is the primary site of lipolysis regulation and prevents the interaction of lipid droplets and lipase by forming a barrier to lipolytic reactions [68]. The phosphorylation of PLIN by PKA eliminates its barrier effect, allowing lipid droplets full contact with ATGL and phosphorylated HSL, which are recruited to the surface of the droplet and initiate lipolysis, promoting thermogenesis. 

In summary, the cAMP-PKA is one of the most intensively studied signaling pathways in mammalian BA thermogenesis under cold stress. Cold stress upregulates the cAMP-PKA signaling pathway in mouse BA, which induces UCP1 expression through p38MAPK, CREB, ATF-2, PGC-1α, etc., to increase thermogenesis and resistance to the cold environment [69] (Figure 1). These signaling pathways play a crucial regulatory role in lipolysis and metabolic energy, and they are the important target for the treatment of metabolic disorders and diseases [70].

### 3.2. AMPK Pathway Regulates BAT Thermogenesis Under Cold Stress

#### 3.2.1. AMPK

AMPK is a serine/threonine protein kinase found in all eukaryotic cells. It plays a pivotal role in regulating various metabolic pathways, and it is highly expressed in both the brain and BAT [71]. AMPK plays a significant role in the differentiation and activation of BAs [72]. AMPK is expressed more and with greater activity in BA compared to other tissues such as the liver, muscle, heart, and WAT [73]. AMPK consists of three subunits: the catalytic α-subunit, the β-subunit containing a glycogen binding domain, and the γ-subunit with regulatory sites for AMP (activation) and ATP (inhibition) [74]. In BA, the α-subunit is the most developed of the subunits and accounts for a major part of AMPK activity [72]. AMPK is activated in a β-adrenergic-dependent manner by stimuli such as cold, exercise, and fasting, thereby modulating the stress response [75]. Mulligan et al. [73] showed that chronic cold exposure results in the increased activation of AMPK in BAT. AMPK therefore plays a pivotal role in BA thermogenesis under cold stress, activated by an elevated AMP/ATP ratio in a complex mechanism that includes the activation of UCP1 expression in BA. Mice lacking AMPK in adipocytes exhibit cold intolerance, demonstrating resistance to β-adrenergic stimulation of BAT, resulting in mitochondrial defects and insulin resistance in BAT [76].

#### 3.2.2. AMPK Pathway Activation of BA Thermogenesis

Low temperature stimulates skin receptors; activates the SNS to release NE, which acts on the hypothalamus to activate BA function; promotes lipolysis and mitochondrial uncoupling through β3-AR binding; increases heat production and energy homeostasis; and resists cold stress [77]. This AMPK cannot be activated in β3-AR KO mice under cold stress [78]. During cold stress, AMPK activates BA thermogenesis by two different mechanisms: (1) hypothalamic AMPK expression and (2) AMPK directly in BA cytosol [79,80]. Partial AMPK activity in BA cytoplasm is caused by a local energy deficiency, whereas hypothalamic AMPK predominantly reflects overall body energy balance and energy levels across various metabolic organs via the SNS. The presence of adenylate kinase in BA results in the activation of AMPK by the secondary messenger cAMP or high AMP and low ATP, which induces NST in BA [81,82]. AMP activates AMPK by allosteric effects in a complex mechanism, and within the range of activation of the α-catalytic subunit, upstream protein phosphorylation of the threonine residue 172 has been observed [83]. Liver kinase B1 (LKB1), also known as serine/threonine kinase 11 (STK11), participates in the regulation of cellular metabolism and proliferation. When the concentration of AMP is elevated and binds to the γ subunit, LKB1 phosphorylates AMPK at Thr172, thereby activating AMPK in BA [84]. Wu et al. [85] found that AMPK deficiency in adipocytes failed to activate downstream genes under cold conditions by β-AR stimulation, thereby inhibiting thermogenesis and energy expenditure, leading to frostbite or even death. Furthermore, AMPK activation leads to the phosphorylation of PGC-1α, a “master regulator” of mitochondrial gene expression that triggers mitochondrial biogenesis and a “master thermogenic factor” in the process of activating BA adaptive thermogenesis [31,86]. Adaptive thermogenesis refers to the process in which PGC-1α and UCP1 are upregulated in BA during cold stress to facilitate thermogenesis [87]. UCP1 is thought to be at the origin of adaptive thermogenesis in BAT, which stimulates cAMP synthesis in BA by uncoupling the respiratory chain to lower the proton gradient and promote mitochondrial thermogenesis [88]. Mice that lack UCP1 in cold environments are highly susceptible to severe hypothermia, likely due to both the direct impairment of uncoupling and the subsequent damage to the electron transport chain [89]. The upregulation of PGC-1α increases UCP1 expression in the inner mitochondria membrane, thereby mediating UCP1-dependent thermogenesis and increasing transcription levels in BA. For example, the cold-inducible PGC-1α interacts with interferon regulatory factor (IRF4) and a range of nuclear receptors, (including PPARg, estrogenassociated receptor (ERR), and thyroid receptor), to enhance UCP1 transcription in BAT [90,91]. Mice are unable to maintain temperature during cold if they have a palmitoyl protein thioesterase-1 (PPT1) deficiency [92]. Upregulating the expression of PGC-1α and UCP1 prompts them to enhance their thermogenic capacity to sustain a constant body temperature in PPT1-KO mice [92]. AMPK also promotes mitochondrial biogenesis through the induction of PGC-1α transcription, which subsequently activates nuclear respiratory factors 1 and 2 (Nrf-1 and Nrf-2), and then mitochondrial transcription factor A (TFAM) is activated [90]. The TFAM expression within the nucleus is crucial for the transcription and replication processes of mitochondrial DNA (mtDNA) [93]. The damage to TFAM or silencing of Nrf-1 resulted in a reduction in mtDNA copy number and impact on mtDNA replication and transcription in BA miochondria [94]. It can be concluded that AMPK is crucial for maintaining mitochondrion content and function in BA, which is essential for energy homeostasis [95] (Figure 2). In conclusion, the AMPK signaling pathway leads directly to BA thermogenesis by promoting the release of fatty acids and the activation of UCP1 under cold stress, which is an important biological response for self-protection and adaptation to cold environments. AMPK is also a pivotal regulator of energy metabolism and stimulates mitochondrial biogenesis in BA, thereby taking part in maintaining energy balance in the body [79].

## 4. Thermogenesis and Energy Balance in Brown Adipose Tissue Under Cold Stress

While energy metabolism refers to the biochemical processes by which the body converts and utilizes energy from nutrients to support cellular functions, energy balance, on the other hand, refers to the equilibrium between energy intake (through food) and energy expenditure (through basal metabolic rate, physical activity, and thermogenesis), determining whether the body gains, maintains, or loses weight [96,97]. The roles of BAT on body weight and energy balance are crucial to homeostasis. In contrast to WAT, BAT is rich in mitochondria and specialized proteins closely tied to energy metabolism, enabling it to consume fat to generate heat for maintaining body temperature [12,98]. The physiological importance of BAT in terms of energy expenditure, insulin sensitivity, weight loss, WAT fibrosis and hepatic steatosis has been emphasized [99]. Cold stress as a potent stimulus affects BAT activity and regulates energy balance [100]. Simcox et al. [101] reported that BAT improves systemic metabolic health by removing metabolites such as lipoproteins, acylcarnitine, and branched-chain amino acids (BCAAs) following exposure to cold. Cold stress increases BAT activity and function, thereby regulating energy balance. Therefore, elevated BAT activity in cold environments plays a role in maintaining body temperature homeostasis and exerts a beneficial influence on body weight and metabolism [102].

### 4.1. Thermogenesis and Energy Balance of Mitochondria in Brown Adipocytes

Mitochondria, the “powerhouse” of the cell, are the principal site of aerobic respiration, which facilitates the oxidation of sugars, fats, and amino acids to synthesize ATP [103]. Mitochondria are the major energy hub in the cell; their mass and quality ensure cellular ATP production to maintain energy homeostasis [104,105]. BAT-mediated thermogenesis exerts a profound impact on systemic metabolism by promoting resting energy consumption, managing systemic glucose levels, and boosting insulin sensitivity [9,106,107].

#### 4.1.1. Mitochondrial Biogenesis Regulates Thermogenesis and Energy Metabolism in Brown Adipocytes

Cold exposure induces BA proliferation and differentiation, and it increases the number of mitochondria in cells [108]. Fatty acid oxidation, thermogenesis, and energy homeostasis in BA allow efficient adaptation to cold [109]. Mitochondrial biogenesis is a sophisticated, multi-stage cellular process involving the transcription and translation of mitochondrial DNA to generate new mitochondria to ensure a sufficient mass of mitochondria. PGC-1α orchestrates the activity of numerous transcription factors involved in mitochondrial biogenesis and function, and it is also a primary regulator of BA mitochondria energy metabolism [110]. PGC-1α activation upregulates the nuclear respiratory factor, estrogen-related receptors (ERR-α, -β and -γ), and TFAM, collectively orchestrating intricate processes involved in mitochondrial biogenesis and energy metabolism [110,111]. Lowell and Spiegelman [112] have demonstrated that cold-induced sympathetic neural activity is responsible for triggering lipolysis and the acute activation of UCP1, as well as the long-term promotion of mitochondrial biogenesis through the induction of PGC-1α. Furthermore, the metabolic sensor AMPK regulates mitochondrial biogenesis via PGC-1α, which increases mitochondria mass and enhances energy metabolism [104,113].

#### 4.1.2. Mitophagy Regulates Thermogenesis and Energy Metabolism in Brown Adipocytes

The quality of mitochondria, which is regulated by mitophagy, impacts energy metabolism. Mitophagy is a type of autophagy that eliminates and degrades damaged mitochondria to ensure mitochondrial integrity and function, thereby maintaining energy homeostasis, and thermogenesis in BAs during cold stress [114,115]. Cold stress causes a large accumulation of oxidative substrates in BA mitochondria, which promotes the production of reactive oxygen species (ROS) in the mitochondria and leads to their accumulation. The ROS accumulation disrupts the mitochondria’s respiratory chain, causing the permeability transition pores to be overly opened; inhibits mitochondrial membrane potentials and ATP production; and leads to mitochondrial dysfunction or even cell death [116]. To ensure mitochondrial quality, the mitochondrial stress sensor PTEN-induced putative kinase 1 (PINK1) and its chaperone E3 ubiquitin ligase (Parkin) are recruited to mitochondrial cristae, initiating mitophagy to facilitate the clearance of damaged mitochondria [117]. Parkin triggers the damaged mitochondria marker to induce mitophagy. Lu et al. [118] demonstrated that cold-induced BA mitophagy is promoted by UCP1 and mediated by PINK1 and Parkin. They found that Pink1 deficiency resulted in insufficient mitophagy flux and reduced energy expenditure in BA activated by cold. Conversely, the inhibition of mitophagy diminished the OXPHOS capacity of mitochondria, indicating that mitophagy is essential for the maintenance of healthy BA mitochondria’s quality in cold stress [118].

#### 4.1.3. Mitochondria Homeostasis Maintains Thermogenesis and Energy Balance in Brown Adipocytes

Energy balance requires that energy intake is equal to energy expenditure. In order to maintain a steady state of mitochondrial mass and quality, the clearance of damaged mitochondria and the generation of new replacements are required for BA thermogenesis and energy balance during cold stress [119,120]. Mitochondrial biogenesis and mitophagy are tightly interconnected, and their balance is essential to maintain metabolic homeostasis and to adapt to energy demands in cold stress [121]. When the balance is disrupted, mitochondrial function is impaired, leading to metabolic disturbances, cellular dysfunction, or even cell death in BAs. PGC-1α serves as a crucial regulator of mitochondrial fatty acid oxidation and biogenesis [122]. PGC-1α activates ERRα, which regulates the expression of key genes that manage energy transfer and ATP synthesis in mitochondria [93]. When PGC-1α was knocked out or silenced, the mitochondrial complexes and mtDNA copy number were reduced, leading to mitochondrial dysfunction and oxidative stress [123]. In certain situations, the over-activation of autophagy can result in abnormal cell death. Rapamycin is a common autophagy inducer, and its excessive activation can further enhance mitophagy activity [124]. Activated SIRT1/PGC-1 enhances mitophagy, leading to excessive degradation components and inhibition of the function of BA [125]. Thus, when mitophagy is greater than mitochondrial biogenesis, it leads to a reduction in mitochondrial mass, impairs mitochondrial function, increases oxidative stress, decreases thermogenic capacity, and causes insufficient energy supply in BA, or even the death of BA [123]. In contrast, when mitochondrial biogenesis outweighs mitophagy, cellular thermogenesis and metabolic activity are enhanced in the short term. But if this imbalance persists, it may lead to mitochondrial dysfunction, an increase in oxidative stress, and even the senescence or death of BAs [126]. Autophagy-related proteins 5 (Atg5) and 7 (Atg7) are important regulatory factors in autophagosome formation. In Atg5 KO mice, the hyperthyroid disrupts its function, resulting in body temperature declining, damaged mitochondria not being removed, and increased ROS, leading to increased oxidative stress [127]. When Atg7 KO inactivates mitophagy, increased mitochondrial numbers and decreased mitochondrial quality lead to a decrease in thermogenesis and metabolism and an increased risk of senescence or lesions in BA [128]. In mammals, the deletion of AMPK or autophagy-activated kinase (ULK1) results in the accumulation of autophagy aptamer p62 and defective mitophagy and metabolic imbalance, leading to a reduction in AMPK-induced thermogenesis and metabolism in BAs [129]. In addition, during the mitochondrial life cycle, fission enables both mitochondrial biogenesis and mitophagy. Mitochondrial fission at the periphery results in the detachment of damaged material into smaller mitochondria, which are targeted for mitophagy, while fission in the central region leads to mitochondrial proliferation; these two processes coordinate to regulate mitochondrial mass and maintain energy metabolism [130]. MAP kinase 3 (MKK3) is an upstream kinase of p38; its deficiency simultaneously activates PGC-1α mediated mitochondrial biogenesis and Parkin/PINK1-mediated mitophagy, leading to the maintenance of a healthy mitochondria network [121]. Similarly, melatonin therapy increases Parkin/PINK1 and PGC-1α/NRF1 expression and enhances mitochondrial biogenesis and mitophagy [131]. The thyroid hormone also promotes mitochondrial biogenesis and mammalian target of rapamycin (MTOR)-mediated mitophagy to increase fatty acid oxidation and mitochondrial respiration, promoting thermogenesis and energy homeostasis in mice BAT in cold environments [127,132]. FUNDC1 is a conserved outer mitochondrial membrane (OMM) protein, a mitophagy receptor [121]. In cold stress, BA produces more mitochondria, and the FUNDC1-mediated mitophagy pathway is activated to maintain mitochondria homeostasis in mice [118,121]. When FUNDC1 was KO, PGC-1α, NRF1, and TFAM mRNA levels in BA were reduced, suggesting that FUNDC1-mediated mitophagy accelerated mitochondrial biogenesis and mitochondria turnover [133]. The deletion of FUNDC1 leads to defects in mitophagy and mitochondrial biogenesis to maintain thermogenesis and energy homeostasis in BAs under cold stress [134]. AMPK is a key regulator of mitochondrial biogenesis and mitophagy in BA under cold stress and regulates energy homeostasis in mice [135]. Thus, these findings suggest that a balance between mitochondrial biogenesis and mitophagy is necessary to maintain mitochondrial function and activate BAT thermogenesis and energy metabolism [127]. Energy balance is achieved by mitochondrial homeostasis, which promotes cell survival and resilience [136] (Figure 3).

### 4.2. AMPK Signaling Pathway Regulates the Energy Balance in Brown Adipose Tissue

AMPK acts as a crucial energy sensor and regulator in BAs [137]. AMPK is involved in the energy sensing and regulatory system and responds to a multitude of physiological, hormonal, and nutritional cues to maintain a balance between ATP production and demand [138]. Mulligan et al. [73] found that α1AMPK activity notably increased after 7 days of cold exposure. Cold stress activates AMPK, promoting fatty acid oxidation, glycogen catabolism, and thermogenesis in BA via elevated NE levels or altered AMP/ATP ratios, enhancing mitochondrial mass, activity, metabolic rate, and energy consumption in response to environmental changes [139]. Mottillo et al. [76] observed that mice lacking AMPK in BAs exhibited cold intolerance and resistance to β-adrenergic stimulation, resulting in a decline in oxidative metabolism. The adaptive thermogenesis and energy expenditure of BA were markedly reduced when stimulated by cold or β-adrenergic stimuli in AMPKα KO mice [85]. AMPK also primarily regulates energy homeostasis by phosphorylating multiple cellular metabolism substrates [140]. AMPK-mediated acetyl-CoA carboxylase (ACC) not only inhibits malonyl CoA but also weakens the inhibitory effect of carnitine palmitoyl transferase 1 (CPT1), promoting fatty acid oxidation and inhibiting fatty acid formation through substrate phosphorylation [141].

The activation of AMPK in BAT involves two distinct mechanisms, i.e., (1) intra-cellular AMPK activation within BA or (2) AMPK downregulation in the hypothalamus [142]. AMPK activation in BAT responds to local energy deficits and cold stimulation and regulates both energy metabolism and cellular thermogenesis, whereas hypothalamic AMPK mainly reflects the whole-body energy homeostasis and enhances energy levels via sympathetic signals flowing to metabolic organs, including the BAT [80]. The main role of AMPK is in maintaining the quality of mitochondria in BA. AMPK mediates mitophagy and the mitochondrial fission factor (MFF) through PGC1α, and unc-51-like ULK1 preserves mitochondria homeostasis and influences energy balance indirectly through mitochondria function in BAs under cold stress [129,143]. In contrast, cold exposure stimulates BA by the flow of SNS from the hypothalamus: after cold exposure, the preoptic area is activated and links to neurons in the paraventricular and ventromedial nuclei of the brain, resulting in the inactivation of AMPK in the hypothalamus; flows through the sympathetic nerves to BAT; and activates BA thermogenesis [144]. Furthermore, these two mechanisms are also interrelated: the inhibition of hypothalamic AMPK activity and the induction of a signaling cascade response result in increased BAT activity and energy expenditure, which is accompanied by NE directly acting on BAs. NE binds to β2-adrenergic (human) or β3-AR (rodent) receptors on the cytomembrane, which activates AMPK, increasing cAMP and phosphorylated PKA and inducing lipolysis in BAs [82]. Chondronikola et al. [106] indicated that β3-AR agonists activate AMPK, which enhances BA metabolic activity in cold-stressed humans [145]. Additionally, AMPK activation facilitates BA differentiation by reducing DNA methylation of the PRDM16 promoter and enhances energy metabolism in BA thermogenesis in vitro [85]. The KO of AMPK inhibited adaptive thermogenesis and energy expenditure in BAT under cold exposure or β3-AR stimulation, leading to symptoms such as impaired glucose tolerance and insulin resistance [85]. Furthermore, estradiol (E2), triiodothyronine (T3), and leptin (Lep) selectively inhibit AMPK in the hypothalamic ventromedial nucleus, activating BAT thermogenesis via the sympathetic nervous system independently of feeding behavior [146,147]. This evidence underscores the crucial role of hormonal regulation in the hypothalamic AMPK-SNS-BAT axis in maintaining energy balance [148] (Figure 4). In summary, BAT increases thermogenesis and regulates energy balance through mitochondrial biogenesis and mitophagy while modulating energy metabolism pathways via the AMPK signaling pathway. Under cold stress, BAT provides the heat and energy required to maintain temperature balance in mammals. The synergistic action of these mechanisms highlights the important physiological regulatory role of BAT in thermogenesis and energy balance [146,149,150,151].

## 5. Research Methods in the Review

This review adopts a systematic literature review approach, primarily utilizing academic databases such as PubMed, Web of Science, and Google Scholar for literature retrieval. The search strategy combined carefully selected keywords and their synonyms, such as “brown adipose tissue”, “cold stress”, “thermogenesis”, “energy metabolism”, as well as relevant animal models (e.g., mice and sheep). The inclusion criteria for the selected literature were as follows: (1) the study must involve the impact of cold stress on brown adipose tissue; (2) it should explicitly discuss the mechanisms of thermogenesis and energy metabolism in brown adipose tissue; (3) the animal experiments should have adequate control groups and reasonable experimental designs. After screening, a total of 167 articles published between 1983 and 2024 were included in this review.

We extracted key data from the selected literature regarding the thermogenic response of brown adipose tissue induced by cold stress, covering metabolic rates, changes in the expression of related genes and proteins, calorie consumption, temperature regulation mechanisms, and cellular pathways associated with cold adaptation. We also conducted an in-depth exploration of the key molecular mechanisms involved in thermogenesis and energy homeostasis in animal brown adipose tissue under cold stress, such as norepinephrine stimulation, UCP1 expression, fatty acid oxidation, and energy balance. By conducting both qualitative and quantitative analyses of the literature, we systematically reviewed and summarized these findings. The quantitative analysis involved extracting numerical data from selected studies, performing statistical evaluations, and conducting meta-analyses where applicable to identify consistent patterns and significant trends. These findings were then integrated into the review and conclusions of this paper, providing a solid theoretical foundation for understanding how animals resist cold environments and increase productivity. Quantitative insights were used to support key arguments, validate hypotheses, and propose practical strategies to elucidate the mechanisms of thermogenesis and energy metabolism in animals under conditions of cold stress.

## 6. Conclusions

Cold stress is often a challenge for the maintenance of homeotherms’ body temperature, leading to growth inhibition, immune dysfunction, and morbidity. This severely constrains the development of the livestock industry in cold parts of the world [152]. Therefore, it is crucial to identify effective treatment and prevention strategies to address the health issues caused by cold stress. BAT possesses remarkable thermogenic capacity and plays a key role in energy homeostasis, maintaining temperature and regulating energy balance in animals [153]. This review suggests that cold stimuli activate the SNS to trigger cAMP-PKA (P38MAPK, CREB, ATF-2, and PGC-1α) and AMPK (AMP/ATP ratio, LKB1, PGC-1α) signaling pathways to regulate UCP1-mediated NST in BA [151]. These two pathways are primarily activated by cold or β-adrenergic agonists, activating BA thermogenesis and energy metabolism. In addition, BA regulates the mass and quality of mitochondria through mitochondrial biogenesis and mitophagy to maintain mitochondria homeostasis and induce thermogenesis and energy balance under cold stress. AMPK induces mitochondrial biogenesis and mitophagy in BA, and its inactivation in the SNS-BAT pathway enhances BAT activity to promote thermogenesis and maintain energy balance. Nevertheless, a comprehensive understanding of the effects of cold stress on BAT energy regulation and other physiological impacts still requires further research. Notwithstanding the aforementioned advances in our understanding of BAT, a number of unanswered questions and challenges remain. Specific gaps include the interactions between BAT thermogenesis and other metabolic processes (such as lipid and glucose metabolism). This is particularly important because diseased or unhealthy cold-stressed mammals often have reduced feed intakes. In addition, there may be significant differences in the response to cold stress and BAT activation between different animal species (such as mice, rats, and livestock), as well as between individuals (such as due to their age, gender, and health status). Research on livestock (such as cattle and sheep) is relatively limited at present, particularly concerning the mechanisms of BAT’s role in cold resistance in these animals. This is despite the fact that hypothermia is a common cause of lamb mortality. Further exploration is also needed on the long-term effects of BAT activation on livestock health and productivity, especially in breeds adapted to cold environments. Research in these areas is crucial for developing effective management strategies to improve livestock welfare and productivity in cold climates. Overall, the studies on the thermogenesis and energy metabolism of BAT under cold stress deepen our understanding of metabolic regulation and offer new perspectives and strategies for the treatment and prevention of metabolic diseases. They also provide a theoretical basis and development prospects for livestock living in cold environments for long periods of time to enable them to survive, improve their feed utilization, reduce mortality, and improve their productivity. Specifically, by activating the thermogenic mechanisms of BAT, the energy metabolic efficiency of livestock in cold environments can be significantly enhanced, reducing energy wastage caused by cold stress. For example, regulating the expression of key proteins such as UCP1 and PGC-1α can optimize the conversion of feed energy into heat energy, thereby improving feed utilization [154]. Additionally, increasing BAT activity helps maintain stable body temperature, reducing disease and mortality rates caused by cold exposure [155]. From a production performance perspective, improving energy metabolic balance can promote growth rate, reproductive efficiency, and overall productivity in livestock, bringing significant economic benefits to the livestock industry in cold regions. In summary, the application of BAT thermogenesis and energy metabolism mechanisms offers a promising avenue for improving livestock performance and welfare in cold environments, and future research should focus on translating these insights into practical interventions to maximize the benefits for livestock producers.

## 7. Future Directions

The response of individual animals to cold varies significantly, so it is necessary to further evaluate BAT levels among different genotypes, including breeds that are adapted to cold. Future research will further explore studies in livestock, comparing BAT functional levels between the different genotypes, with the ultimate objective of breeding sheep with higher BAT content around the spine.

## Figures and Tables

**Figure 1 ijms-26-03233-f001:**
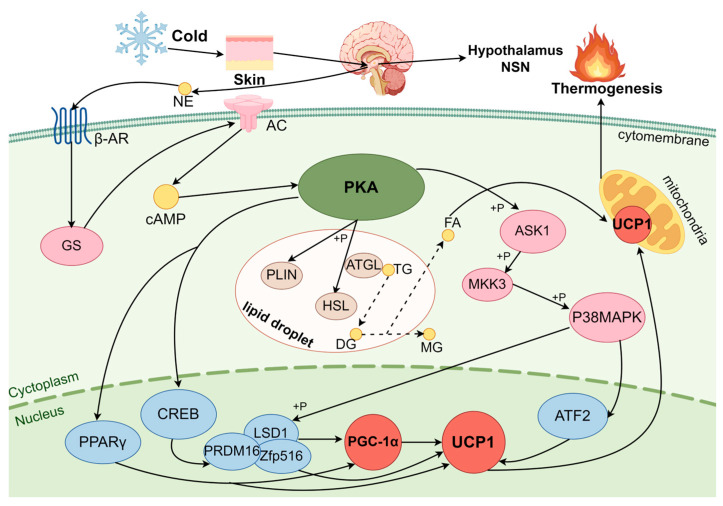
cAMP-PKA signaling pathway in brown adipocyte thermogenesis. Cold stimuli are transmitted to the hypothalamus through the skin, indirectly releasing norepinephrine (NE) to activate the cAMP-protein kinase A (PKA) signaling pathway via the β3-adrenergic receptor (β3-AR). PKA phosphorylation activates factors upstream of the activation of the peroxisome-proliferator-activated receptor gamma coactivator-1α (PGC-1α) in the nucleus and induces the expression of PGC-1α. This activates intranuclear uncoupling protein (UCP1) and acts on mitochondria for thermogenesis. The PKA-dependent activation of p38 mitogen-activated protein kinase (MAPK) activates intranuclear PGC-1α, which induces the expression of UCP1 and promotes thermogenesis in BA. PKA promotes lipolysis within lipid droplets, increases the release of free fatty acids (FFAs), and utilizes UCP1 in mitochondria to regulate BA thermogenesis. Adenylate cyclase (AC), cAMP response-origin binding protein (CREB), peroxisome-proliferator-activated receptor γ (PPARγ), lysine-specific demethylase1 (LSD1), positive regulatory domain-containing protein (PRDM16), MAP kinase kinase (MKK3), zinc finger protein (Zfp516), transcription factor 2 (ATF2), hormone-sensitive lipase (HSL), adipose triglyceride lipase (ATGL), and unesterified fatty acids (FA).

**Figure 2 ijms-26-03233-f002:**
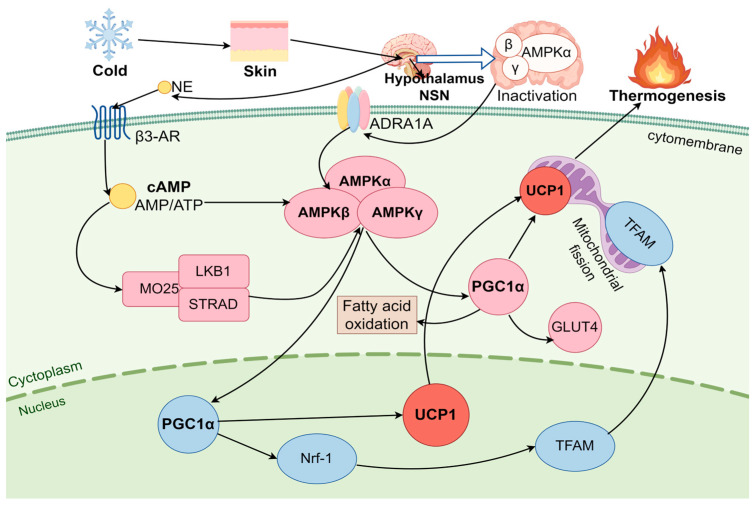
AMPK signaling pathway in brown adipocyte thermogenesis. Cold stimuli are delivered to the hypothalamus through the skin and activate the AMP-activated protein kinase (AMPK) signaling pathway directly via BA surface adrenergic receptor α1A (ADRA1A) or indirectly by releasing norepinephrine (NE), which activates the signal AMPK pathway via the adrenergic receptor (β3-AR). AMPK activation by upstream multimers (Liver kinase B1 LKB1, Mouse protein-25 MO25, STRAD) induces the phosphorylation of peroxisome-proliferator-activated receptor gamma coactivator-1α (PGC-1α) in the nucleus, increasing UCP1 transcription and promoting thermogenesis. Upregulating AMPK directly activates intracellular PGC-1α, which induces UCP1 expression and enhances thermogenesis. AMPK activates nuclear respiratory factor 1 (Nrf-1) by inducing PGC-1α transcription, upregulates mitochondrial transcription factor A (TFAM) expression, and increases mitochondrial DNA transcription and replication. Glucose transporter 4 (GLUT4).

**Figure 3 ijms-26-03233-f003:**
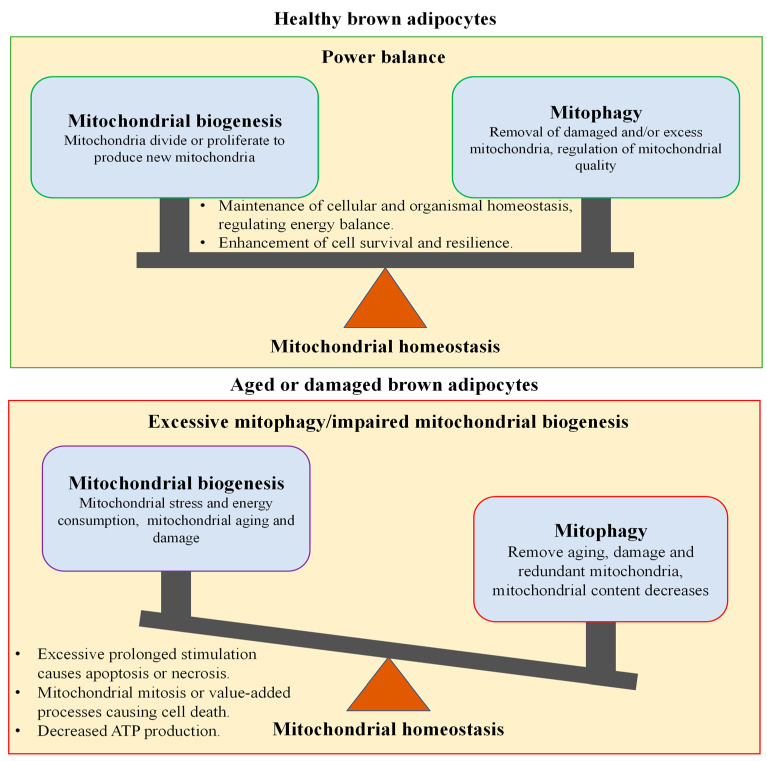
Mitochondrial biology and mitophagy coordinate organismal energy homeostasis [104,121]. (1) Healthy BA mitochondrial homeostasis and maintaining energy balance. (2) Aged/damaged mitochondria, unable to maintain mitochondrial homeostasis, result in defective thermogenesis and energy homeostasis. Mitophagy is greater than mitochondrial biogenesis, and mitophagy is excessive/damaged mitochondrial biogenesis. (3) Mitochondrial biogenesis is greater than mitophagy, and mitochondrial biogenesis is excessive/damaged mitophagy.

**Figure 4 ijms-26-03233-f004:**
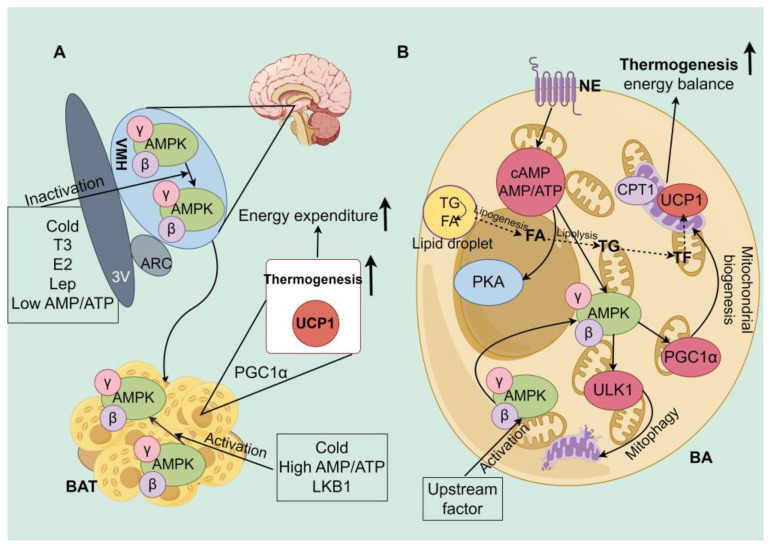
AMPK mediates brown adipocyte thermogenesis and energy regulation [74,151]. A: Hypothalamic AMPK-SNS-BAT axis. BAT is activated by inhibiting hypothalamic AMPK, and SNS transmits signals from AMPK inactivation to BAT to upregulate lipolysis and promote thermogenesis and energy regulation. B: BA-AMPK axis. Norepinephrine (NE) released via the SNS binds to the β3-adrenergic receptor (β3-AR), activating cAMP-PKA and AMPK, which ultimately increases lipolysis and thermogenesis. AMPK also regulates energy homeostasis via mitochondrial biogenesis and mitophagy homeostasis. The activation of AMPK leads to an increase in the uptake of triglyceride (TG)-derived nonesterified fatty acids (NEFA) from lipoproteins and inhibits carnitine palmitoyl transferase 1 (CPT1) in mitochondria, enhances FA transport to mitochondria, and promotes lipolysis for thermogenesis. Estradiol (E2), triiodothyronine (T3), leptin (Lep), an unc-51-like autophagy kinase 1 (ULK1).

## Data Availability

No data were used for the research described in this article.

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
