# Peer review of "Thermogenesis and Energy Metabolism in Brown Adipose Tissue in Animals Experiencing Cold Stress"

_ijms, 2025, doi:10.3390/ijms26073233_

Round 1
Reviewer 1 Report
Comments and Suggestions for Authors
Review:
Overall:
The paper presents strong scientific content; however, grammatical and stylistic improvements are recommended for conciseness and clarity. Additionally, the author should provide more insight into the practical significance of covering the mechanisms of BAT thermogenesis under cold stress.
Positive attributes:
1. The authors prepared a nice review summarizing cold exposure effects from 163 articles published between 1983 and 2024.
2. The authors aimed to connect mechanistic knowledge of BAT thermogenesis with real world applications. Clear focus on thermogenesis and energy metabolism in brown adipose tissue (BAT) under cold stress and good logical flow starting with role of BAT in thermogenesis to signaling pathways and mitochondrial homeostasis and ending with potential applications (more on that later).
Negative attributes:
Minor:
1. Line 11, “Critical in necessitating" is redundant
2. Line 18, line 117, line 165, line 184, line 315, BA (Brown adipocytes) is uncountable, so use of article “the” before BA should be avoided. E.g. BA instead of “the BA”
3. Line 565, line 566, Same issue with BAT and the BAT
4. Line 54, line 98, Non-shivering thermogenesis (NST) have been referred to as NTS
5.“To understand the mechanisms of thermogenesis and energy metabolism in BA under cold stress, we described the processes that cold stress activates BAT thermogenesis and energy metabolism regulation." Sentence unclear.
6. Table 1: Reference source not found
7. Line 535, “By performing qualitative and quantitative analysis of the literature, we summarized these findings, providing a solid theoretical foundation for animals to resist cold environments and enhance productivity.” Authors should add how the quantitative analysis was done and how they used the findings in this paper.
8. Line 540, morbidity is uncountable, “morbidity” instead of “the morbidity”
9. Line 119, line 281, line 298, line 310, line 346, line 350, line 357, line 361, line 368, line 401, line 404, line 417, line 422, line 425,line 426, line 435, line 500, line 550 "Mitochondria" is plural, but should be "mitochondrial" when used as an adjective.
10.Line 560, Livestock instead of “livestocks”, livestock is uncountable noun.
Major:
1. Lack of novelty statement: A review generally shares new insight while summarizing findings in a field. It is unclear the novel insight(s) the authors are sharing.
Line 21: "This research identifies potential therapeutic targets for developing new strategies to treat metabolic diseases, as well as providing theoretical support for optimizing cold stress response strategies and improving energy homeostasis in mammals."
No component protein of the mechanisms described or illustrated have been identified/suggested as a drug target and no new potential strategy offered.
2. Insufficient discussion between the mechanisms described with livestock benefits:
Line 559 “It also provides a theoretical basis and development prospects for livestocks living in cold environments for long periods of time to enable them to withstand cold environments, improve the feed utilization, reduce the mortality and improve the productivity.”
How the authors view the application of knowing the mechanism of BAT thermogenesis in surviving cold, improving feed utilization, reduction of mortality and improvement of productivity is not obvious and can use some more discussion with specific ideas and benefits.
3. What specific gaps remain?
Line 553, "Nevertheless, a comprehensive understanding of the effects of cold stress in BAT energy regulation and other physiological impacts still requires further research."
A little discussion on what specific gaps remain would be useful.
Reviewer 2 Report
Comments and Suggestions for Authors
Dear Authors,
I really appreciate your tremendous effort you did for this thorough review, to understand the mechanisms of thermogenesis, to provide the theoretical support for optimizing cold stress response strategies and improving energy homeostasis in mammals and practically the animal farm reproduction and production. You reached the point the manuscript is to be accepted for publication, just after minor correction
